# Numerical Analysis of the Effect of Liquid Water during Switching Mode for Unitised Regenerative Proton Exchange Membrane Fuel Cell

**DOI:** 10.3390/membranes13040391

**Published:** 2023-03-29

**Authors:** Hock Chin Low, Bee Huah Lim

**Affiliations:** Fuel Cell Institute, Universiti Kebangsaan Malaysia, Bangi 43600, Selangor, Malaysia

**Keywords:** URPEMFC, two-phase flow, flow field, computational fluid dynamics

## Abstract

As unitised regenerative proton exchange membrane fuel cell (URPEMFC) is progressing in terms of its performance, more emphasis should be placed on the understanding of the interaction between multiphase reactants and products and its effect during the switching mode. A 3D transient computational fluid dynamics model was utilised in this study to simulate the supply of liquid water into the flow field when the system switched from fuel cell mode to electrolyser mode. Different water velocities were investigated to identify their effect on the transport behaviour under parallel, serpentine, and symmetry flow fields. From the simulation results, 0.5 m·s^−1^ water velocity was the best-performing parameter in achieving optimal distribution. Among different flow field configurations, the serpentine design achieved the best flow distribution due to its single-channel model. Modification and refinement in the form of flow field geometric structure can be performed to further improve the water transportation behaviour in URPEMFC.

## 1. Introduction

Carbon emissions from the burning of fossil fuels or coal for the purpose of human development contribute to roughly two-thirds of all greenhouse gases produced, which are currently the leading cause of the worsening climate change. On the other hand, renewable energy has been envisioned as a promising solution to realise the zero-emission power generation method. Simultaneously, the establishment of the Paris Agreement by the United Nations aiming to decarbonise society has spurred the development of hydrogen technology. Unitised regenerative proton exchange membrane fuel cell (URPEMFC), a fairly novel renewable energy device, serves as an efficient electrochemical system integrating fuel cell and electrolysis operation into a single cell to generate power and store energy by utilising hydrogen. Hence, this fuel cell can play a major role in the rise of sustainable energy solutions. 

The operation of URPEMFC involves producing hydrogen during electrolyser mode (EM) and generating electricity during fuel cell mode (FCM). The system is typically designed to operate in a constant electrode mode, where hydrogen oxidation and oxygen evolution reactions take place at the anode, while oxygen reduction and hydrogen evolution reactions take place at the cathode [1]. This electrode configuration has several advantages over the constant-gas mode, such as the ability to utilise the existing material development and technology from the conventional proton exchange membrane fuel cell (PEMFC) and electrolyser system. However, one of the major disadvantages involves the mixing of liquid and gas reactants in the flow field in both operation modes, which requires a purging process during mode switching [2]. Therefore, a major challenge in this system design is the intricacy of water and gas management within the flow field. Excessive water content from FCM in the flow field limits the gas permeability on the gas diffusion layer (GDL), and further degradation of GDL limits the flow path of liquid water, causing severe water flooding problems [3]. The residual water content from EM also affects the mass transport of hydrogen gas during FCM at the anode [4]. Additionally, insufficient water content within the cell also causes membrane dehydration, which leads to accelerated degradation in the system [4]. In this regard, many studies have been conducted to optimise the purging process during mode switching, such as the purging time, flow rate, purging gas temperature and pre-switching gas purge [5,6,7]. 

Another aspect to consider during the optimisation of the mode switching of URPEMFC is thermal management. An even and constant operating temperature is necessary for the system to perform at its intended performance. However, the switching of reactants could potentially reduce the cell temperature from the incoming lower temperature liquid or gas, causing interruption to the electrochemical reaction rate occurring in the system. This then leads to a decrease in cell performance and hydrogen generation until it reaches a steady state [8]. Aside from the individual cell, the difference in temperature within a stack has also been seen to affect the two-phase flow behaviour [9]. Thus, effective mode switching can be achieved with optimised operating parameters throughout the system to ensure a smooth and optimal transition. Apart from that, the duration for the reactants to switch instantaneously can affects the electrochemical reaction rate [10]. Hence, a well-optimised reactant switching process improves the voltage response in the system, which ensure a stable transition.

At present, most studies focus on the gas purging process, specifically during the conversion from EM to FCM. Alternatively, the switch from FCM to EM also faces some difficulties, such as the conversion to a higher operating voltage and the possibility of water starvation or uneven distribution of liquid reactants due to gas blockage. Many studies have been conducted to understand the fluid transportation of two-phase flow within a flow field through experimental studies or numerical analyses [5,11,12,13,14]. However, these methods do not provide much clarity in the optimisation process in terms of the fluid flow characteristics in the flow field. Computational fluid dynamics (CFD) has the ability to demonstrate the dynamics of liquid-gas movement and the flow characteristics in different types of flow fields while experimenting with various parameters. This study, therefore, aims to understand the phenomena by utilising the volume of fluid (VOF) model in CFD to accurately simulate the interaction between liquid and gas reactants in a typical URPEMFC flow field. By utilising CFD simulation, the flow field design process to ensure well-distributed fluid will be more efficient. This is also crucial for optimising the mode switching in URPEMFC.

The use of VOF has shown to be an insightful and reliable method in simulating two-phase flow operation, as reported in various studies. Cao et al. investigated the water removal capability of gas channels with different cross-sectional dimensions in PEMFC [15]. The flow pattern of water bulk and droplets was demonstrated with the interaction with external air forces. The VOF method was also used to study the reactant transport capability of a 3D fine mesh flow field with air intake for a better water removal rate [16].in The flow characteristics of such two-phase flow interaction were shown through the simulation, indicating a decrease in water coverage on the GDL surface and an increase in the water removal rate. Therefore, in this study, multiple flow field designs were investigated using the VOF method to understand their effect on the dynamic interaction between liquid and gas while experimenting with different inlet water velocities. The geometry of the flow field is expected to be essential for an efficient distribution of reactants, including a smooth transition between URPEMFC modes. Thus, a CFD approach was employed to capture the transport behaviour of liquid water using the VOF model.

## 2. Methods

This study utilised parallel and serpentine flow fields as the more conventionally studied designs and symmetry flow fields inspired by the bio-inspired flow field. Figure 1 shows the 3D computational domain with respect to each of the flow field designs. The flow path was set to travel through the domain in the manner of a Z-shape at an upright orientation with a single inlet and outlet at each side. The active area of parallel and serpentine flow fields is about 11 × 11 mm^2^, whereas the active area of the symmetry flow field is 15 × 15 mm^2^. The dimension of the flow channel has a cross-sectional area of 1.0 × 1.0 mm^2^. The parallel model configuration has straight channels bifurcating perpendicularly from the inlet flow path and merging together leading to the outlet. Next, the serpentine model features a simple single-channel design with multiple sharp U-shaped turning points. Finally, the symmetry model, which is a flow field design created for this study, has a flow channel angling towards the inlet at 45°, directing the flow towards the outlet.

Table 1 presents the operating parameters for the CFD simulation. The effect of temperature and pressure was excluded from the study. Therefore, the water was supplied at room temperature and the atmospheric pressure was set. The transport behaviour of liquid water and gas was investigated at the targeted velocity range, which was comparatively slow. Several assumptions were made in the modelling of the simulation: (1) based on the dimensions of the flow channel and the fluid velocities during the test, the Reynolds number of liquid water is well within the laminar region; (2) as there are no external heat sources, isothermal condition is assumed; (3) the fluid is also considered to be incompressible with the fixed flow field volume; and (4) phase change is excluded from the study due to the fast transport of liquid water across the domain within the timeframe and the absence of heat source. The basic boundary conditions of the model are shown in Figure 1. The flow field channel is enclosed by upper, lower, and side walls with the default material.

The numerical simulation to investigate the fluid behaviour of liquid water was carried out using the commercial CFD software ANSYS Fluent R1, 2022 (ANSYS, Inc., ANSYS Drive, Canonsburg, PA, USA) with the Volume of Fluid (VOF) model. The equations necessary to capture the transport process of the two-phase flow interaction are listed below. The continuity equation is given in Equation (1), where ρ is the density and $\vec{v}$ represents velocity vector:(1)$$\frac{\partial \rho }{\partial t}+\nabla \cdot \left(\rho \vec{v}\right)=0$$

Equation (2) describes the momentum equation, where $\vec{g}$ is gravity, μ is the viscosity of the mixture, and ρ is the density, which can be defined by Equations (3) and (4). The term $S_{m}$ represents surface tension:(2)$$\frac{\partial }{\partial t}\left(\rho \vec{v}\right)+\nabla \cdot \left(\rho \vec{v}\cdot \vec{v}\right)=-\nabla P+\mu \nabla \cdot \left(\nabla \vec{v}+\nabla {\vec{v}}^{T}\right)+\rho \vec{g}+S_{m}$$
(3)$$\rho =\varepsilon_{liquid}\rho_{liquid}+\varepsilon_{gas}\rho_{gas}$$
(4)$$\mu =\varepsilon_{liquid}\mu_{liquid}+\varepsilon_{gas}\mu_{gas}$$
where ε is the volume fraction of liquid and gas, respectively, with the relationship the sum of two phases volume fraction is equals to 1. The term $S_{m}$ in the momentum equation also put into consideration of the surface tension effect and smoothing of the interface between the two phases, is defined by Equation (5), where α is surface tension coefficient and K is the surface curvature:(5)$$S_{m}=\alpha K\frac{\rho \nabla \varepsilon_{liquid}}{\frac{1}{2}\left(\rho_{liquid}+\rho_{gas}\right)}$$

## 3. Results and Discussion

The dynamic interaction of two-phase flow was simulated through CFD to investigate and visualise the transport behaviour and characteristics between liquid water and gas within the bipolar plate’s flow field during mode switching. Parallel, serpentine, and symmetry flow field designs are considered in this study with a single inlet and outlet at the opposite end. The effect of different inlet water velocities was investigated through different flow field configurations to observe the flow distribution and optimise for an efficient transition. The colour in the figures displayed in this section represent the volume fraction of liquid water, with red representing the highest volume fraction and blue the absence of water in the zone.

### 3.1. Water Velocity

From the transient simulation results, the inlet water velocity significantly affects the behaviour of the flow distribution within each flow field. Across the three different inlet water velocities investigated, 0.5 m·s^−1^ showed the best flow uniformity across all flow field designs. Figure 2 presents the distribution performance in terms of the volume fraction of liquid water after reaching a steady state throughout different flow fields. The main finding is that the parallel flow field struggled to achieve a uniform flow distribution in all channels due to its geometric structure. Even though the water velocity of 0.5 m·s^−1^ in the parallel flow field displayed unsatisfactory flow distribution, it worsens at a lower velocity, as shown in Figure 3a. Another behaviour observed in the parallel flow field as the inlet velocity increased is the increasing amount of back-flow in the channels. This could be due to the increasing pressure that has built up on the last channel, which forces water back into the previous channels. A potential solution to this is capturing the fluid velocity across the domain through further CFD simulation and optimising the pressure drop in the flow field. 

At a lower water velocity of 0.25 m·s^−1^ (Figure 3), all flow field designs experienced difficulty in forming substantial liquid film across all channels, thus leading to poor flow distribution. This result is in line with other studies, where a higher inlet velocity is preferable to increase the efficiency of the mass transfer of liquid reactant and also to achieve a better gas removal rate [9,17]. At an increased velocity of 0.75 m·s^−1^, no noticeable improvement was observed at a steady state for both symmetry and serpentine configurations. As shown in Figure 4f, symmetry flow field showing similar flow pattern as Figure 2c. Furthermore, the parallel flow field achieved a steady state at 0.15 s, as shown in Figure 5c. Then moving towards the serpentine flow field shown in Figure 6f, again similar flow pattern was observed. A higher water velocity resulted in the build-up of liquid film across the flow field at a higher rate leading to a steady state, but the flow distribution was significantly affected by the flow field geometric structure. As the temperature of the fluid is excluded from this simulation study, it is to be noted that the effect of water velocity on the interaction of the two-phase flow has been reported to have a correlation with operating temperature [18]. Therefore, there is a further improvement on the current flow field design with optimisation in both the flow rate of liquid water and temperature simultaneously.

### 3.2. Flow Field Configuration

Aside from the different inlet water velocities investigated, the geometric structure of the flow field pattern directly influences how liquid water is distributed across the domain effectively, as determined from the simulation. As shown in Figure 4, the 45° orientation of the flow channel in the symmetry flow field introduced a lot of vortices throughout the domain. The pressure differential formed past the bipolar plate ribs that were angled towards the fluid flow direction affected its flow regime at multiple spots. This phenomenon is expected to slow down the liquid reactant distribution and potentially cause the water starvation region. Another observation made for the symmetry flow field is that the middle channel closer to the inlet provides an easy straightforward flow path towards the outlet, creating a lower pressure drop compared to other channels. This causes the channels that are offset to left and right from the middle channel struggle to form sufficient liquid films.

As for the parallel flow field shown in Figure 5, at each bifurcation point on top, the liquid flow tends to follow the easier path rather than the channel perpendicular to the inlet flow path direction. This leads to the rapid fluid build-up towards the channel of the end, causing water starvation at the channels closer to the inlet even after reaching a steady state at 0.15 s. Similar observation can be seen from multiple experimental studies, where the distribution of liquid water varies in different channel arms due to different pressure drop characteristics across each channel [14]. No significant improvement was observed at the low inlet velocity. This is a common phenomenon observed in the parallel flow field, where the resistance can vary between each channel, especially in the case of two-phase flow, which causes poor water management [19]. An improvement can be achieved by modifying the parallel flow field geometry [20,21] or optimising the manifold design to improve the distribution [22]. 

On the other hand, the serpentine flow field obtained well-distributed fluid, as seen in Figure 6. This is expected for a single-channel flow field, as it has been reported to have a better distribution than multiple channels [23]. However, the multiple U-shaped turning points could be seen dispersing the fluid flow, as it wraps around and slows down the flow rate, causing a delayed steady state within the flow field. This flow characteristic is expected for the serpentine flow field high pressure drop design from the strong adhesion effect with the channel wall and multiple sharp turning points [24]. The vortex created at the tip of the channel rib is even more pronounced in the serpentine flow field as compared to the symmetry flow field. This phenomenon is also observed in other experimental studies, where liquid water bulk is separated in a serpentine flow field with air pockets throughout [13]. Therefore, based on simulation findings, it is expected that a higher water velocity will be beneficial in ensuring uniform flow distribution. Nevertheless, once the liquid is transported throughout the domain arriving at a steady state at around 0.3 s, the serpentine flow field typically achieves an optimal distribution with the single-channel design. However, the effect of the air pocket formed throughout the channel wall at 0.3 s should be further studied.

## 4. Conclusions

In this study, the fluid transport behaviour of the mode switching from FCM to EM was visualised through the numerical simulation of the two-phase flow using the VOF method in ANSYS Fluent. During this transition, the supply of liquid water in the anode kickstarts the electrolysis operation and also serves to purge the reactants and products from FCM operation to ensure the absence of any impurities. Additionally, the fluid distribution across the flow field also significantly affects the transition efficiency and system performance. Therefore, this study utilises 3D transient CFD simulation to study the distribution of liquid water in a gas-filled flow field and observe the behaviour of the two phases in parallel, serpentine, and symmetry flow fields. The simulation results show that the design of the flow field greatly impacts the effectiveness of the two-phase flow distribution, specifically the geometric design and pressure drop characteristics. Among the inlet water velocities investigated, 0.5 m·s^−1^ achieved notably better liquid water distribution. As for the flow field design, the parallel flow field showed the worst distribution performance. The serpentine flow field achieved good flow distribution for the most part due to its single-channel design. The symmetry flow field struggled to achieve a uniform flow distribution, but with optimised water velocity, an optimal flow distribution can be obtained. This study successfully demonstrated the flow pattern of two-phase flow in the flow field using the VOF method. Future studies can utilise this simulation method to evaluate new flow field designs or improve the geometric design of existing flow fields. As the mass transport study has been a focus for more than 10 years in PEMFC, this simulation is expected to provide a more realistic approach to simulate the transport through GDL [25]. A more comprehensive model can also be produced by introducing the hydrophobicity or hydrophilicity of the material, the temperature of the fluid, and the pressure differential between the system and the reactant supply storage in order to conduct a more comprehensive optimisation of the mode switching process.

## Figures and Tables

**Figure 1 membranes-13-00391-f001:**
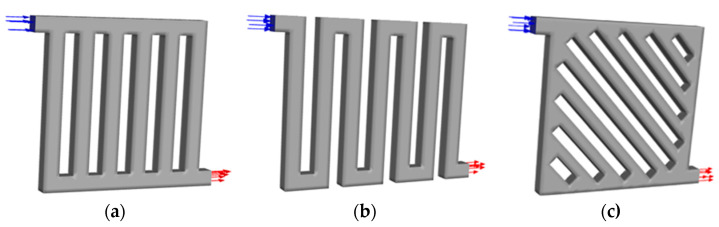
Computational domain of flow field models with arrow indicating the flow direction: (**a**) parallel; (**b**) serpentine; and (**c**) symmetry.

**Figure 2 membranes-13-00391-f002:**
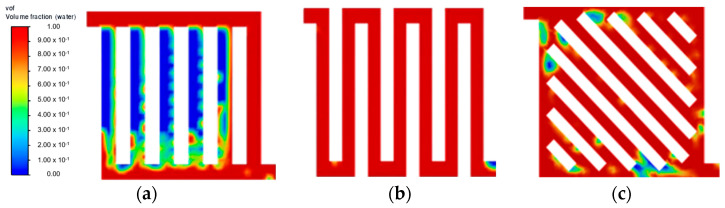
Steady state flow distribution at velocity = 0.50 m·s^−1^: (**a**) parallel; (**b**) serpentine; and (**c**) symmetry.

**Figure 3 membranes-13-00391-f003:**
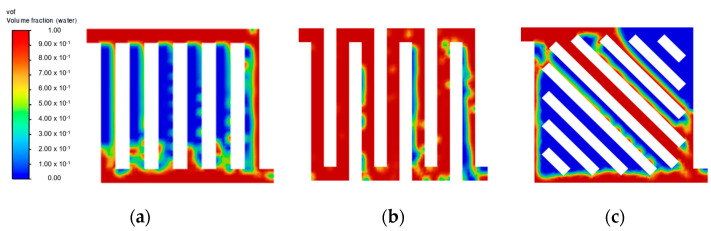
Steady state flow distribution at velocity = 0.25 m·s^−1^: (**a**) parallel; (**b**) serpentine; and (**c**) symmetry.

**Figure 4 membranes-13-00391-f004:**
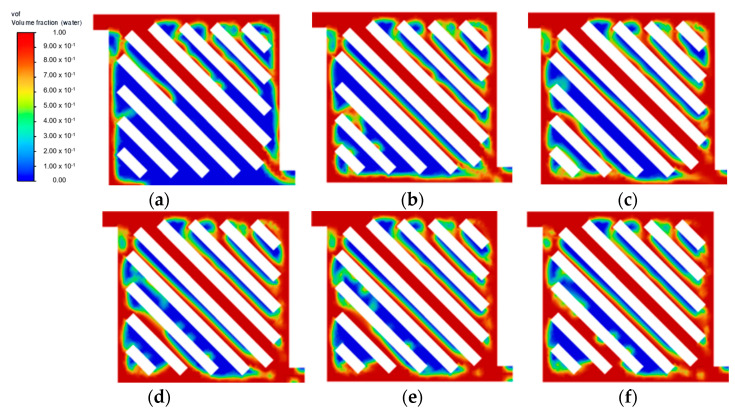
Flow distribution with velocity = 0.75 m·s^−1^: (**a**) 0.05 s; (**b**) 0.10 s; (**c**) 0.15 s; (**d**) 0.20 s; (**e**) 0.25 s; and (**f**) 0.30 s.

**Figure 5 membranes-13-00391-f005:**
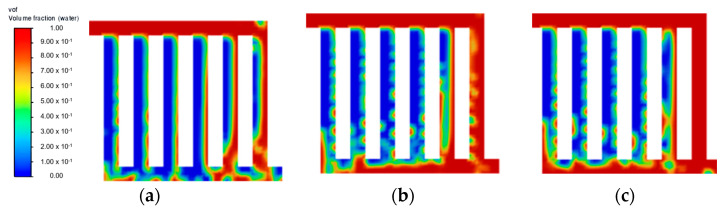
Parallel flow field with velocity = 0.75 m·s^−1^ (**a**) 0.05 s; (**b**) 0.10 s; and (**c**) 0.15 s.

**Figure 6 membranes-13-00391-f006:**
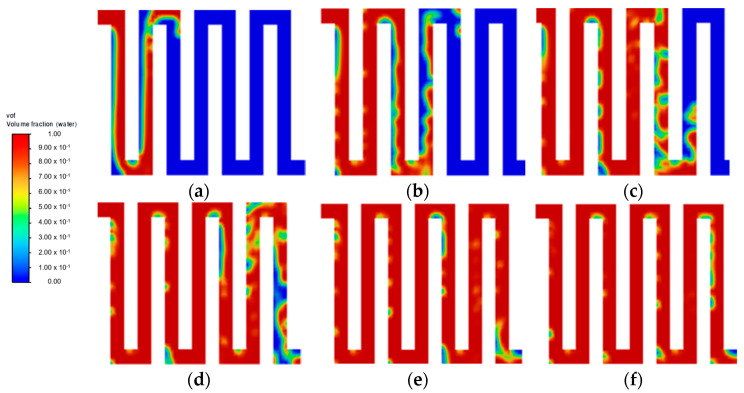
Serpentine flow field with velocity = 0.75 m·s^−1^ (**a**) 0.05 s; (**b**) 0.10 s; (**c**) 0.15 s; (**d**) 0.20 s; (**e**) 0.25 s; and (**f**) 0.30 s.

**Table 1 membranes-13-00391-t001:** Operating parameters.

Parameters	Value
Operating temperature (K)	293.15
Operating pressure (atm)	1.0
Gravity (m·s^−2^)	9.81
Water-liquid Velocity (m·s^−1^)	0.25/0.5/0.75

## Data Availability

Not applicable.

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
