# Peer review of "Numerical Analysis of the Effect of Liquid Water during Switching Mode for Unitised Regenerative Proton Exchange Membrane Fuel Cell"

_membranes, 2023, doi:10.3390/membranes13040391_

Round 1

Reviewer 1 Report

1.The model should be described in more detail. Did the author verify the simulation results?

2.The pictures in the text should give the corresponding volume fraction range for the different colors.

3.Figure 3 should give a cloud map of the flow fraction of the other two flow fields.

4.In the section of introduction, I think there are other related references in this research field as follow:

1) .Lattice Boltzmann simulation of the structural degradation of a gas diffusion layer for a proton exchange membrane fuel cell. Journal of Power Sources, 2023, 556: 232452.

5.Some data formats need to be corrected, such as 0.5 m/s (page 4 line 159 and page 5 line 162).

6.The meanings of some symbols in Equation (2) are not given.

7.The conclusion section should be written even more concise.

Reviewer 2 Report

It is considered that the document is almost ready, however, it is recommended that the authors add the following references to complement the manuscript with respect to the specific studies that have been carried out in proton exchange membrane fuel cells.

https://doi.org/10.1016/j.ijhydene.2014.02.156

  https://doi.org/10.3390/su11236682

https://doi.org/10.1016/j.jpowsour.2013.06.078

Before being accepted, authors must attend to these minor revisions.

Reviewer 3 Report

Comments on Manuscript.

Manuscript number:  membranes-2185422

The article entitled "Numerical Analysis of the effect of liquid water during switching mode for unitized regenerative proton exchange membrane fuel cell " has been reviewed. My comments and recommendations about the recent article can be found in below. The paper can be accepted.

In General;                           

* The subject of current investigation and method carried out are within the scope of membranes.

* The topic of switching mode represent a crucial component of the PEM fuel cell stack is quite interesting and the obtained results would be very useful.

* The quality of English grammar is quite satisfactory.

* The obtained results can be very useful for scientific community.

As per my opinion I have drawn the following views given as below:-

1.     Originality:- The running text in present article has shown sufficient novelty and excellent interest to warrant for final publication.

2.     Results and discussion:- The way of explanation in this part is found very much strong, and I am fully agreed and results presented in this article are in enough satisfaction.

3.     Conclusion:- Excellent

Overall, the presentation of work is quite well and satisfactory. The outcomes of the study will add new information to the scientific literature and will be very helpful for the forthcoming workers of the material science.

Author Response

Thank you for going through the manuscript and acknowledging our work.